# Applicability of Goethite/Reduced Graphene Oxide Nanocomposites to Remove Lead from Wastewater

**DOI:** 10.3390/nano9111580

**Published:** 2019-11-07

**Authors:** Franklin Gordon-Nuñez, Katherine Vaca-Escobar, Milton Villacís-García, Lenys Fernández, Alexis Debut, María Belén Aldás-Sandoval, Patricio J. Espinoza-Montero

**Affiliations:** 1Escuela de Ciencias Químicas, Pontificia Universidad Católica del Ecuador, Av. 12 de Octubre 1076, Apartado, 17-01-2184 Quito, Ecuador; frank-bep@hotmail.com (F.G.-N.); lmfernandez@puce.edu.ec (L.F.); 2Departamento de Ingeniería Civil y Ambiental, Escuela Politécnica Nacional, Ladrón de Guevara E11·253, PO·Box 17-01-2759 Quito, Ecuador; maria.aldas@epn.edu.ec; 3Facultad de Ciencias Químicas, Universidad Central del Ecuador, Francisco Viteri s/n y Gato Sobral, PO Box Ciudadela Universitaria, 170129 Quito, Ecuador; mhvillacis@uce.edu.ec; 4Centro de Nanociencia y Nanotecnología, Universidad de las Fuerzas Armadas ESPE, Av. Gral. Rumiñahui s/n, P.O. Box 171-5-231B Sangolquí, Ecuador; apdebut@espe.edu.ec

**Keywords:** adsorption, goethite, graphene oxide, lead

## Abstract

Lead ion in drinking water is one of the most dangerous metals. It affects several systems, such as the nervous, gastrointestinal, reproductive, renal, and cardiovascular systems. Adsorption process is used as a technology that can solve this problem through suitable composites. The adsorption of lead (Pb(II)) on graphene oxide (GO) and on two goethite (α-FeOOH)/reduced graphene oxide (rGO) composites (composite 1: 0.10 g GO: 22.22 g α-FeOOH and composite 2: 0.10 g GO: 5.56 g α-FeOOH), in aqueous medium, was studied. The GO was synthesized from a commercial pencil lead. Composites 1 and 2 were prepared from GO and ferrous sulfate. The GO and both composites were characterized by using scanning electron microscopy (SEM), scanning transmission electron microscopy (STEM), Raman spectroscopy, X-ray diffraction (XRD), Fourier-transform infrared spectroscopy (FTIR), and dynamic light scattering (DLS). The adsorption capacity of Pb(II) on the GO and both composites was evaluated through adsorption isotherms. Composite 1 presented a significant agglomeration of α-FeOOH nanorods on the reduced graphene oxide layers. Meanwhile, composite 2 exhibited a more uniform distribution of nanorods. The adsorption of Pb(II) on the three adsorbents fits the Langmuir isotherm, with an adsorption capacity of 277.78 mg/g for composite 2200 mg/g for GO and 138.89 mg/g for composite 1. Composite 2 emerged as a highly efficient alternative to purify water contaminated with Pb(II).

## 1. Introduction

Heavy metals have been the major pollutants in the last decades, around the world. Due to their persistence, bioaccumulation, biomagnification, and toxicity, they represent a threat to aquatic life, and especially to human health [1]. Lead ion [Pb(II)] is the thermodynamically more stable lead species and is one of the most hazardous heavy metals; it affects the nervous, gastrointestinal, reproductive, renal, and cardiovascular systems [2]. In addition, it represents a threat to the environment, especially near mining or smelting zones [2]. Although adsorption is the most employed process to remove heavy metals, the use of inefficient adsorbent materials for wastewater treatment could be the reason why Pb(II) is found in concentrations over the permitted limits in drinking water (maximum contaminated level, MCL = 0.015 mg/L) [3]. A clear example is activated charcoal, which has a lower adsorption capacity compared to other materials and a relatively high cost [4]. As a result, in recent years, efforts to find alternative adsorbents have increased. Due to their low cost and wide availability, natural materials such as clays and metal oxides are being studied in more detail [4]. Among the adsorbent materials, GO is an alternative that has been recently tested for the removal of heavy metals, and with good results [5]. GO’s ability to form hybrids with other materials is remarkable. Its functional groups (hydroxyl, carboxyl, and epoxide) allow it to participate in many interactions. In addition, this material can be used as a substrate for the growth of metal oxides and hydroxides, improving its adsorbent properties [6]. On the other hand, goethite (α-FeOOH) is one of the most studied iron oxyhydroxides. It is a mesoporous material with high potential for the adsorption of contaminants, due to its low solubility and easy laboratory synthesis [7]. This material has shown good results in regard to the absorption of metals such as lead, arsenic, copper, chromium, and zinc, overcoming the adsorption capacity of materials such as activated charcoal, hematite and zeolite [8,9]. Additionally, the use of GO for the formation of hybrids has been demonstrated to remove contaminants from wastewater [7,8,9,10]. For these reasons, the applicability of goethite (α-FeOOH)/reduced graphene oxide (rGO) nanocomposites was proposed to remove Pb(II) from wastewater. The composites synthesis was previously reported by Xu et al. [11]; they obtained four composites with different mass ratios, but, for this work, they chose the two most different mass ratios reported (composite 1: 0.10 g GO: 22.22 g α-FeOOH and composite 2: 0.10 g GO: 5.56 g α-FeOOH), in order to better understand each component’s contribution to the lead adsorption process. In this way, it could be applied for the wastewater treatment from different sources. It is important to emphasize that the results can be extrapolated for the adsorption of other heavy metals.

## 2. Materials and Methods

### 2.1. Materials and Reagents

Graphite was obtained from a commercial 9B graphite pencil (Cretacolor, Hirm, Austria). Ultrapure water was obtained by using the water purification system (Thermo Fisher Scientific B-Pure, Waltham, MA, USA). Potassium permanganate (KMnO_4_) and lead nitrate II [Pb(NO_3_)_2_] were purchased from PanReac AppliChem (PanReac AppliChem, Barcelona, Spain). It was used polyvinylidene difluoride (PVDF) membranes of 0.45 μm pore size and nitrocellulose membranes of 0.02 μm pore size (Merck Millipore, Burlington, MA, USA). Ferrous sulfate heptahydrate (FeSO_4_∙7H_2_O), 37% (*w*/*w*) hydrochloric acid (HCl), 35% (*V*/*V*) hydrogen peroxide (H_2_O_2_), 65% (*w*/*w*) nitric acid (HNO_3_), 75% (*V*/*V*) phosphoric acid (H_3_PO_4_), sodium hydroxide (NaOH), and 96% (*V*/*V*) sulfuric acid (H_2_SO_4_) were purchased from Merck KGaA (Merck KGaA, Darmstadt, Germany). The GO and the nanocomposites were dried in a stove (Thermo Fisher Scientific, Waltham, MA, USA).

### 2.2. Graphite Cleaning

The graphite was obtained from a commercial 9B graphite pencil, because it is the one with a higher percentage of graphite and less clay (kaolinite) compared with other commercial series. Since the graphite could contain impurities, an acid–base cleaning was performed. First, 3 g of the pencil lead was ground to a fine powder. The graphite was immersed in 150 mL of a 20% (*w*/*w*) NaOH solution and stirred for three hours. The solid was filtered through a PVDF membrane (pore size: 0.45 µm) and stirred in 800 mL of ultrapure water (18.2 MΩ). Then, the graphite suspension was filtered, 150 mL of a 2 M HCl solution was added, and it was then stirred for one hour. The solid was filtered and stirred again in 800 mL of ultrapure water for half an hour. Finally, the graphite was filtered through a PVDF membrane (pore size: 0.45 μm) and dried at 104 °C for 15 h.

### 2.3. Graphene Oxide Synthesis

The GO was synthesized following the methodology described by Marcano et al. [12]: 1 g of graphite powder was added to 133 mL of a solution with a 9:1 ratio of 96% (*V*/*V*) H_2_SO_4_ and 75% (*V*/*V*) H_3_PO_4_. The mixture was agitated while it was placed in an ice bath. Then, 6 g of KMnO_4_ was added slowly, producing an exothermic reaction that reached 35 °C, which was controlled by the ice bath. Next, the mixture was heated and stirred at 50 °C for 12 h. After the reaction was completed, the solution was cooled to room temperature and added to 133 mL of ice. Then, 2 mL of 35% (*V*/*V*) H_2_O_2_ was added. The mixture was filtered through a polyester fiber to remove non-oxidized graphite remains. Then, the filtrate was centrifuged for 4 h at 4000 rpm. The supernatant was discarded, and the solid phase (GO) was washed twice with 150 mL of ultrapure water, to eliminate residual H_2_SO_4_. Also, the solid phase (GO) was washed two more times with 150 mL of HCl 35% (*V*/*V*), to remove metal ions. Finally, it was washed twice with 150 mL of ethanol, to raise the pH. Afterward, the mixture was sonicated for one hour. Finally, it was vacuum-dried at 50 °C for 2 days.

### 2.4. The α-FeOOH/rGO Composites Synthesis

The composites were prepared according to the method proposed by Xu et al. [11].

Composite 1: A suspension of 500 mL of GO in water with a concentration of 0.20 mg/mL was prepared. In the suspension, 69.50 g of FeSO_4_∙7H_2_O was dissolved, followed by the addition of 125 mL of ethylene glycol (EG). The solution was stirred for half an hour and refluxed at 100 °C for three hours. After the formation of the composite, the solution was cooled at room temperature and filtered through a PVDF membrane (pore size: 0.45 µm). The composite was washed twice with 800 mL of ultrapure water and twice with 800 mL of ethanol. Finally, the product was vacuum-dried for 12 h at 60 °C.

Composite 2: A suspension of 125 mL of GO in water with a concentration of 0.80 mg/mL was prepared. In the suspension, 17.38 g of FeSO_4_∙7H_2_O was dissolved, followed by the addition of 31.25 mL of EG. The solution was stirred for half an hour and refluxed at 100 °C for three hours. After the formation of the composite, the solution was cooled at room temperature and filtered through a PVDF membrane (pore size: 0.45 µm). The washing and drying processes were carried out in the same way as with composite 1.

### 2.5. Characterization

The SEM and STEM images of the compounds were obtained with a TESCAN microscope (MIRA 3). The Raman spectra were determined with a HORIBA Scientific LabRAM Evolution spectrometer, with a laser of 532 nm wavelength, at a power of 100 mW. The XRD spectra of the compounds were obtained with a Panalytical diffractometer (Empirian), at a wavelength of 1.54 Å and a power of 45 kV. The FTIR spectra were generated using a PERKIN ELMER Spotlight 200 spectrometer, with a wave number range from 4000 to 450 cm^−1^. Finally, the surface charge of the compounds (Z potential) was determined using a DLS 90 Plus Particle Size Analyzer BROOKHAVE, at different pH values.

### 2.6. Optimum pH Value Determination

For the optimum pH value determination, the chemical speciation program MINEQL was used to obtain the Pb(II) species distribution diagram as a function of pH for the studied conditions. This curve, together with the Z-potential graphs of the compounds, allowed us to find a pH that was not so alkaline that the Pb(II) would precipitate, nor so acidic that there would be competition with the H^+^ ions (optimum pH). The source of Pb(II) was Pb(NO_3_)_2_. (The optimum pH range determined was over 5, so all of the experiments were developed at pH 5.)

### 2.7. Adsorption Kinetics

For adsorption kinetics, jar-test equipment (7790-901B) with 100 mL plastic containers was used. For each container, a 50 mL solution with a concentration of 80 mg/L of Pb(II) and 5 mg of GO, for a concentration of 0.10 g/L of the adsorbent, was prepared. The pH value was adjusted to the optimum (pH 5), with 0.01 M NaOH and 0.01 M HNO_3_ solutions. The samples were kept under agitation at 250 rpm. Also, the pH values were monitored and regulated. A sample was taken from each container at different time intervals. The samples were filtered through nitrocellulose membranes (pore size: 0.02 μm). The same procedure was carried out for composites 1 and 2. Finally, the Pb(II) concentration of each sample was determined by Atomic Absorption Spectroscopy (AAS), using a SHIMADZU spectroscope, Model AA-6300.

### 2.8. Adsorption Isotherms

For adsorption isotherms, in five containers, 50 mL of solutions with different Pb(II) concentration were prepared: 80, 160, 240, 320, and 400 mg/L, respectively. Next, 5 mg of GO was added to each container for a concentration of 0.10 g/L of the adsorbent. In addition, a 50 mL solution with a concentration of 40 mg/L of Pb(II), which was used as a blank, was prepared. Subsequently, the pH of the solutions was adjusted to the optimum (pH 5) with acid or alkaline solutions. An agitation of 250 rpm was maintained, and the pH value was regulated periodically. After reaching equilibrium time, a 20 mL sample was taken from each container. The samples were filtered through nitrocellulose membranes of 0.02 μm pore size. This same procedure was executed for the composites. Finally, the Pb(II) concentration of each sample was determined by AAS.

To understand the adsorption process, it is necessary to analyze the isotherms. The most commonly used models are Langmuir (Equation (1)) and Freundlich (Equation (2)) [13,14].
(1)$$q_{e}=\frac{q_{max}KC_{e}}{1+\left(KC_{e}\right)}$$
where $q_{e}$ is the amount of adsorbate sorbed per unit of adsorbent weight (mg/g), $C_{e}$ is the equilibrium concentration (mg/L), *K* is the constant that relates the adsorbent affinity by the adsorbate (L/mg), and $q_{max}$ is the maximum amount of adsorbate sorbed per unit of adsorbent weight (mg/g).
(2)$$q_{e}=kC_{e}^{1/n}$$
where *k* is the constant that indicates the adsorption capacity of the adsorbent [L/mg] and n is the constant that indicates the adsorption intensity (dimensionless).

## 3. Results and Discussion

### 3.1. Graphite Cleaning

Figure 1a shows the Raman spectrum of graphite before cleaning. The D peak, which represents the presence of impurities and/or defects in the graphite structure, has a considerably high intensity compared to the G peak, whose presence is due to the stretching of the pairs of carbon atoms sp^2^, indicating the presence of an impurity. In addition, the 2D band, second order of the D peak, is not well defined, and this confirms the contamination of the graphite [15,16].

The Raman spectrum of graphite after cleaning shown in Figure 1b reveals a low intensity D peak compared to the G band, and a well-defined 2D peak, indicating that the initial material was successfully cleaned [16].

### 3.2. Graphene Oxide Synthesis

The color change of the mixture during the GO synthesis is an indicator of a correct procedure (see Figure 2) [17]. The first color variation occurred after the addition of KMnO_4_ to the graphite suspension. The mixture changed from transparent (Figure 2a) to a dark green (Figure 2b). According to Dreyer et al. [18], this color is due to the formation of Mn_2_O_7_, which is the species responsible for the graphite oxidation. Once the Mn_2_O_7_ reacted with the graphite domains, the color of the mixture became purple–brown, as shown in Figure 2c, due to the release of MnO_2_ [19]. After the oxidation process and the addition of ice to the mixture, it turned a dark purple color due to the residual KMnO_4_ (Figure 2d) [17].

After adding the H_2_O_2_, the mixture acquired a yellow color (Figure 2e) due to the reduction of residual KMnO_4_ and MnO_2_ into colorless soluble salts, which revealed the true color of GO [19]. According to Dimiev et al. [20], the synthesized GO possesses domains from six to seven cyclic chains of carbon. Such a characteristic gives GO a strong tonality. The bright yellow color indicates that the GO absorbs in a narrow region of the blue light of the visible spectrum, suggesting that the aromatic domains have a similar size. Finally, after washing with water, the GO acquired a dark brown color (Figure 2f). According to Dimiev et al. [20], this characteristic is caused by GO light absorption in a wider region of the visible spectrum. Its reaction with water could cause the formation of extra C=C bonds and the formation of carbonyl groups, which are strong chromophores.

### 3.3. The α-FeOOH/rGO Composites Synthesis

As in the GO synthesis, the color change during the preparation of the composites is an indicator of a correct synthesis [21]. Figure 3 reveals the color variation of GO when it was reduced. Figure 3a shows the brown color obtained by GO in water, while Figure 3b indicates the dark tone that the compound acquired after reflux. According to Xu et al. [11], the GO reduction occurs in two stages. In the first one, the Fe(II) ions are adsorbed by GO functional groups. In the second, the formation of α-FeOOH crystal nuclei occurs during the reflux of the mixture. The majority of GO functional groups are removed by the action of Fe(II) ions, resulting in a reestablishment of the π-π conjugation of the compound. This means that a dramatic increase in the absorption intensity through the UV-Vis region occurred, conferring a black color to the composites [21]. It is important to note that the change from brown to black occurred both in composite 1 and composite 2.

### 3.4. Characterizations

#### 3.4.1. SEM Analysis

Figure 4 shows the SEM images of (a) graphite, (b) GO, (c) composite 1, and (d) composite 2. In the case of graphite, the material is composed of smooth surface flakes [22]. For the GO, the flakes are no longer visible; instead, a smooth, homogeneous, and uniform surface was observed [23]. The compound has a paper-like appearance, with several wrinkles on its surface [24]. On the other hand, the SEM image of composite 1 shows that the GO sheets are completely covered by the α-FeOOH forming an irregular three-dimensional structure. This result suggests that the GO was successfully reduced and that the α-FeOOH grew on its surface. In addition, the SEM image of composite 2 indicates the growth of α-FeOOH on the GO surface [10,12]. It is important to mention that the three-dimensional structure of the composites means a higher specific area in comparison to the smooth surface of the GO; this is a favorable factor in the adsorption of pollutants [11]. EDX analysis of graphite, GO, rGO, and composites confirm the chemical reaction as described in Section 2. As can be seen in Table 1, the normalized mass percent of C decreases from nearly 71% (graphite) to 16% for reduced GO. As expected, the Fe mass percent increases for reduced GO, as Fe is one the chemical element of the reaction.

#### 3.4.2. STEM Analysis

Figure 5 indicates the STEM images of (**a**) graphite, (**b**) GO, (**c**) composite 1, and (**d**) composite 2. In graphite’s case, Figure 5a shows a material flake, with its typical layered structure [24]. On the other hand, Figure 5b reveals the separation of graphene sheets to form the smooth structure of the GO, which presents wrinkles on its surface [25]. In the case of the composites, the images reveal the growth of α-FeOOH nuclei in the form of nanorods on the rGO surface [10]. However, for composite 1, the nanorods appear agglomerated on the surface and cover it completely, and for composite 2, the nanorods are distributed in a more uniform and spaced way. Composite 1 synthesis employed a low GO concentration, which means a smaller specific area when it was suspended. Therefore, the agglomeration of α-FeOOH on the rGO surface is probably due the low specific area in which the mineral nuclei could grow. Meanwhile, for composite 2, the higher concentration of GO allowed a more spaced growth of α-FeOOH [11].

#### 3.4.3. Raman Spectroscopy

Figure 6 shows the Raman spectrum of (**a**) graphite and (**b**) GO. The composites were not characterized by this technique, because the equipment laser oxidized the samples and transformed them into another compound. This process was corroborated with techniques such as XRD and FTIR. In Figure 6a, graphite has a low-intensity D peak compared to the G peak (ID/IG = 0.04), which is associated with nanocrystalline carbon. In addition, the I2D/IG ratio = 0.21 shows the existence of multilayers [16]. On the contrary, Figure 6b reveals that the D and G bands have almost the same intensity (ID/IG = 0.97), which confirms a greater disorder of the basal planes of the GO, due the functional groups that are found on the surface of each layer. This indicates that the material was correctly oxidized (peak ~1345.04 cm^−1^ is associated with amorphous carbon materials) [26]. In addition to D and G bands, the material has a wider and less intense 2D band. These characteristics corroborate the disorder of the material layers and indicate that GO is formed by multilayers, in accordance with I2D/IG = 0.09 [27].

#### 3.4.4. XRD Analysis

Figure 7 shows the XRD patterns of (a) graphite, (b) GO, (c) composite 1, and (d) composite 2. In the case of graphite, a peak at 2θ = 26.46° is observed, which corresponds to the reflection of the plane (002) characteristic of graphene layers [10]. In addition, using Bragg’s law, it was determined that the spacing between graphene layers (d-space) is 3.37 Å [28]. For the GO, a band can be visualized at 2θ = 9.77°, which belongs to the reflection of the plane (001) characteristic of the GO [29]. The d-space of the GO is 9.05 Å. The increase in distance between the GO layers is due to the presence of functional groups in the basal planes of the material [11]. 

In the composite 1 spectrum, several peaks at 2θ = 17.80°, 21.20°, 26.55°, 33.29°, 34.78°, 36.58°, 39.99°, 41.16°, 53.27°, 58.98°, 61.41°, 63.97°, and 71.61° can be observed, which correspond to the reflection planes (020), (110), (120), (130), (021), (111), (121), (140), (221), (151), (002), (061), and (042), respectively. These results are consistent with the JCPDS card (No. 29–713) of pure α-FeOOH, which shows the growth of that compound over the rGO layers [30]. The composite 2 spectrum presents the same peaks as composite 1. However, it is important to note that, in the composite 1 spectrum, the characteristic diffraction peak of graphite (2θ = 26.46°) cannot be visualized. This is because each rGO layer was completely covered on both sides by α-FeOOH [11]. For composite 2, in addition to the peaks of α-FeOOH, a low intensity peak can be visualized at 2θ = 25.86°. This is probably due to the characteristic spectrum of rGO, because the α-FeOOH was uniformly distributed but did not saturate or generate an agglomeration on the surface.

#### 3.4.5. FTIR Analysis

Figure 8 shows FTIR spectra of (a) graphite, (b) GO, (c) composite 1, and (d) composite 2. The graphite spectrum indicates the presence of a single significant peak at ~3241 cm^−1^, corresponding to the O–H stretching vibrations of water molecules deposited on the surface of the material [24]. On the other hand, the GO spectrum shows several peaks: The band that goes from ~3700 cm^−1^ to 3000 cm^−1^ is due to the O–H stretching vibrations of water molecules, as well as the O–H stretching vibrations of hydroxyl and carboxyl groups [26], while the band at ~1417 cm^−1^ is caused by the O–H bending vibrations of the mentioned groups [27]. The band at ~1736 cm^−1^ is attributed to the C=O stretching vibrations of carboxyl and carbonyl groups [31]. On the other hand, the peak at ~1627 cm^−1^ is due to the C=C skeletal vibrations of graphite [11], while the bands at ~1230 cm^−1^ and ~1067 cm^−1^ are caused by the C–O–C stretching vibrations of epoxy groups [32,33]. The presence of all these functional groups demonstrated an appropriate oxidation of the material.

For composite 1, Figure 8c shows a narrower band compared to GO at ~3250 cm^−1^ that is caused by the stretching vibrations of O–H atoms in α-FeOOH structure [34]. The bands at ~877 cm^−1^ and ~781 cm^−1^ are attributed to the Fe–O–H bending vibrations in the material [11]. Meanwhile, the peak at ~673 cm^−1^ is due to the Fe–O stretching vibrations in α-FeOOH [10]. It is important to note that the bands at ~3250, ~877, and ~781 cm^−1^ have similar intensities, which is typical for pure α-FeOOH [11]. Additionally, it is important to remember that composite 1 presented a significant agglomeration of α-FeOOH over the rGO layers. As a result, the FTIR spectrum may correspond to the α-FeOOH nanorods that covered the rGO surface only.

For composite 2, as in composite 1, the following observations were made: a band at ~3256 cm^−1^ due to the O–H stretching vibrations, two bands at ~891 cm^−1^ and ~774 cm^−1^ attributed to the Fe–O–H bending vibrations, and a peak at ~653 cm^−1^ by the Fe–O stretching vibrations [10,12,34]. In the case of composite 2, the intensity of the peaks at ~891 cm^−1^ and ~774 cm^−1^ is lower than the intensity of the peak at ~3256 cm^−1^, which would indicate a higher proportion of O–H bonds in the material. Due to this peculiarity, the material might not have been completely dry. However, in the studies carried out by Xu et al. [11], the α-FeOOH/rGO composites also presented this characteristic. Therefore, there is the possibility that the intense band at ~3256 cm^−1^ is due to the structure of the material or to the uniform distribution of the nanorods in the rGO.

#### 3.4.6. Z-Potential Determination through DLS Analysis

Figure 9 indicates the surface charge of the GO and the composites at different pH values. GO has a negative surface charge in the pH range between 3 and 8. According to Dimiev et al. [35], this characteristic is due to the creation of vinylogous carboxyl groups by the reaction of the GO functional groups with water. In the analyzed pH range, the carboxyl groups are ionized in vinylogous carboxylates, which are responsible for the negative charge of the GO. 

On the other hand, since the composites were coated by α-FeOOH, it is very likely that their surface charge was caused by the structure of the mineral. Both composites (1 and 2) have a negative surface charge at a pH higher than ~3.30 and a positive charge at a lower pH. According to Aquino et al. [36], the hydroxyl groups of α-FeOOH can protonate or deprotonate in solution, as a function of pH. At very low pH values, the OH groups adsorb H^+^ ions, causing a positive charge on the material surface, while at higher pH values, the OH^–^ groups of α-FeOOH release H^+^ ions, resulting in a negative charge on the surface.

### 3.5. Optimum pH Value Determination

Figure 10 shows the distribution diagram of Pb(II) species for a concentration of 400 mg/L of Pb(II), which was used in batch experiments. In the pH range between 5.5 and 11, the Pb(II) can form three solid compounds: [Pb_3_(OH)_2_O(CO_3_)_2_], Pb(OH)_2_, and PbCO_3_, because their saturation indices are equal or very close to zero. Therefore, the pH value used was 5, making sure that the total Pb(II) was in solution. It is important to mention that, at this pH value, the compounds have a negative surface charge (Figure 9); therefore, the Pb(II) adsorption was expected to be successful.

### 3.6. Adsorption Kinetics

Figure 11 shows the equilibrium time needed for the Pb(II) adsorption into GO and composites. For the GO, the equilibrium time was reached in 30 min. According to Zhang et al. [37], the material is extremely hydrophilic, so it disperses easily in solution. This peculiarity allows immediate internal transport, external transport, and adsorption of Pb(II) onto GO. In the case of composites 1 and 2, the equilibrium time was reached at 360 min. The reduction of the GO by FeSO_4_∙7H_2_O explains the disappearance of its functional groups and, therefore, of its rapid dispersion in water [38]. However, the composites’ equilibrium time was reached in a few hours compared to pure α-FeOOH, which has reported equilibrium times of weeks [39]. This particularity might be due to its nanometric size, which speeds up the internal transport of the pollutant.

### 3.7. Adsorption Isotherms

Figure 12 shows the non-linearized isotherms for the adsorption of Pb(II) onto GO and composites. For the three compounds, at low concentrations of Pb(II), the amount adsorbed was significant (higher slope). However, as pollutant concentration increases, the active sites begin to saturate, so the compounds adsorb a lower amount of Pb(II) (lower slope). This behavior is called L-type isotherm, which can be interpreted by both Langmuir and Freundlich isotherms [40].

The higher slope of the GO at low concentrations indicates a greater affinity between the Pb(II) and the compound, probably due to the good dispersion capacity of the GO in water. However, this characteristic does not necessarily mean that the GO has a higher adsorption capacity than the composites [41]. In addition, the GO reaches its maximum saturation point because the curve becomes asymptotic, which does not happen with the composites. Hence, it is necessary to linearize the curves to know the maximum adsorption capacity of the three compounds.

The parameters corresponding to the Langmuir and Freundlich isotherms for the adsorption of Pb(II) onto GO and composites are shown in Table 2. Based on the correlation coefficient values, it can be noted that the experimental data fit the Langmuir isotherm. Therefore, the compounds have a homogeneous adsorbent surface. The maximum adsorption capacity (q_max_) was found to be 200 mg/g for the GO, 138.89 mg/g for composite 1 and 277.78 mg/g for composite 2, which would indicate that the three materials are applicable in the adsorption of Pb(II), in comparison to certain adsorbents tested years ago, such as activated carbon (47.62 mg/g), carbon nanotubes (49.94 mg/g), bentonite (20 mg/g), and even pure α-FeOOH (11.04 mg/g) [42,43,44,45].

As could be seen in the SEM and STEM images of the composites, the growth of α-FeOOH nanorods on the rGO surface suggests a higher specific surface area and therefore a greater space where Pb(II) can be adsorbed. Thanks to this characteristic, both composites should have a higher adsorption capacity (q_max_) than the GO; however, this did not happen with composite 1. According to Xu et al. [11], the agglomeration of α-FeOOH reduces the specific surface area of the composite, which means a decrease in its q_max_. On the other hand, the contribution of the main and secondary faces of the nanorods should be considered. It is likely that the agglomeration of nanorods in composite 1 means that the solution could only be in contact with the secondary faces of the α-FeOOH, as is suggested at Figure 13a. Thus, the main faces, which would be together, would not contribute in the adsorption of Pb(II). In the case of composite 2, the nanorods appeared more separated from each other, as we suggest in Figure 13b, allowing it to have a greater adsorption capacity due the contribution of both the secondary and main faces.

The excellent results found for the remission of Pb(II) from contaminated wastewater led us to propose this material for other metal ions, such as Cd(II), Cu(II), As(V), Zn(II), etc. In our research group, we are currently testing adsorption by using Cd(II) and As(V), with excellent results that will be published in due course.

## 4. Conclusions

The characterization of the materials through SEM, STEM, Raman spectroscopy, XRD, and FTIR allowed us to verify the proper synthesis of the GO and the composites. Using the STEM images, we could differentiate the morphology between the two composites. In the case of composite 1, a significant agglomeration of α-FeOOH nanorods was observed, while for composite 2, the nanorods were evenly distributed and more spaced with each other. The adsorption isotherms of Pb(II) onto GO and composites at a pH value of 5 were adjusted to the Langmuir model. It was determined that composite 2 was the material with the highest adsorption capacity (277.78 mg/g), followed by the GO (200 mg/g) and composite 1 (138.89 mg/g). The three synthesized compounds presented a good adsorption capacity for Pb(II), surpassing the capacity of adsorbents used worldwide, such as activated carbon (47.62 mg/g), low-cost materials like wheat bran (64 mg/g), and even goethite (11.04 mg/g) [42,45,46].

## Figures and Tables

**Figure 1 nanomaterials-09-01580-f001:**
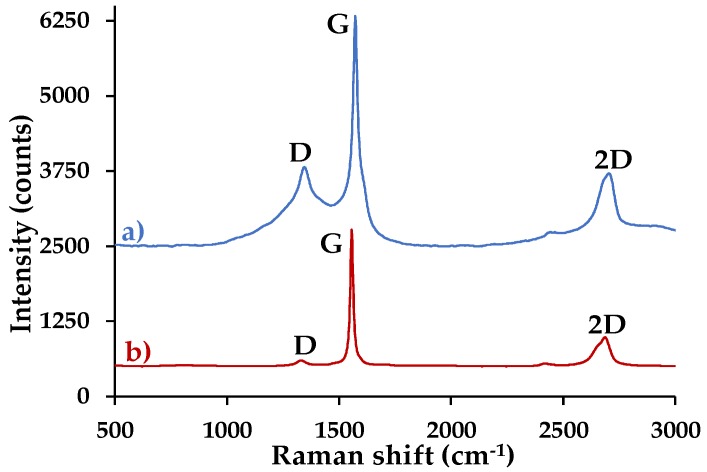
Raman spectrum of graphite: (**a**) before cleaning and (**b**) after cleaning.

**Figure 2 nanomaterials-09-01580-f002:**
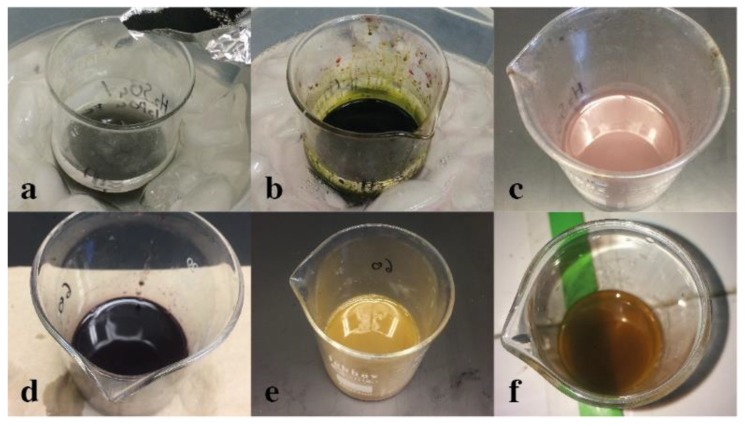
Color change of the mixture during the synthesis of GO. (**a**) Colorless of the suspension of graphite in acid; (**b**) green color due to the formation of Mn_2_O_7_ when KMnO_4_ was added to the mixture; (**c**) purple–brown mixture due to the formation of MnO_2_; (**d**) dark purple color due to the residual KMnO_4_ in presence of water; (**e**) H_2_O_2_ produced a yellow color; (**f**) the GO reaction with water generated a dark brown color.

**Figure 3 nanomaterials-09-01580-f003:**
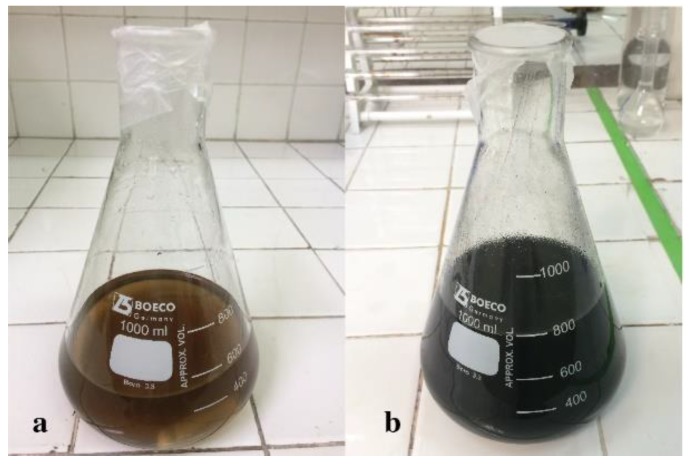
Color change of the mixture during the synthesis of the composites. (**a**) The suspension of GO in water had a brown tone; (**b**) the mixture turned black after reflux.

**Figure 4 nanomaterials-09-01580-f004:**
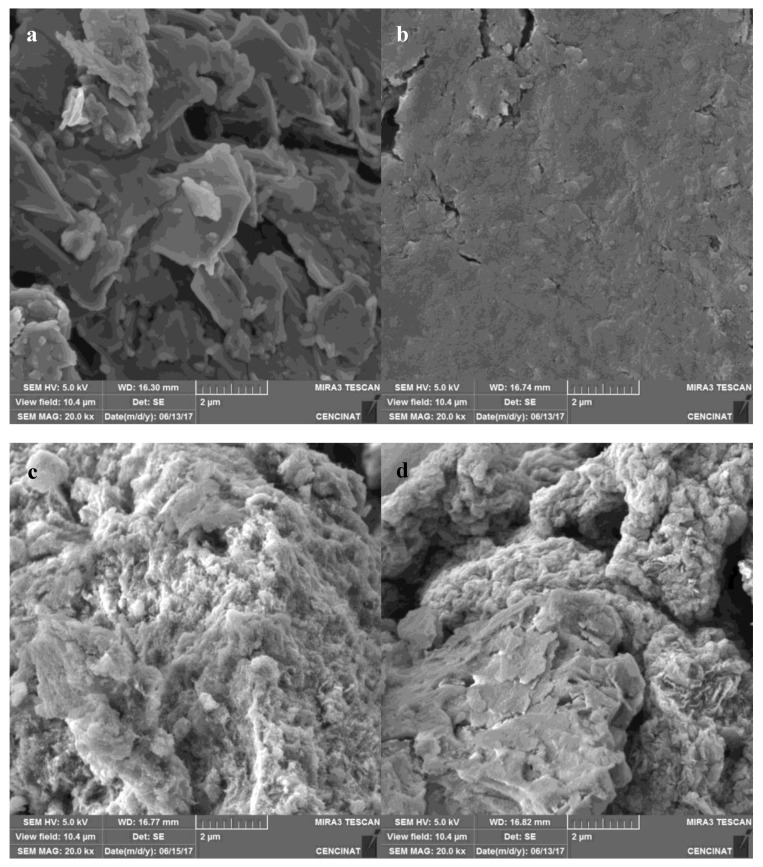
SEM images of (**a**) graphite, (**b**) GO, (**c**) α-FeOOH/rGO 1, and (**d**) α-FeOOH/rGO 2.

**Figure 5 nanomaterials-09-01580-f005:**
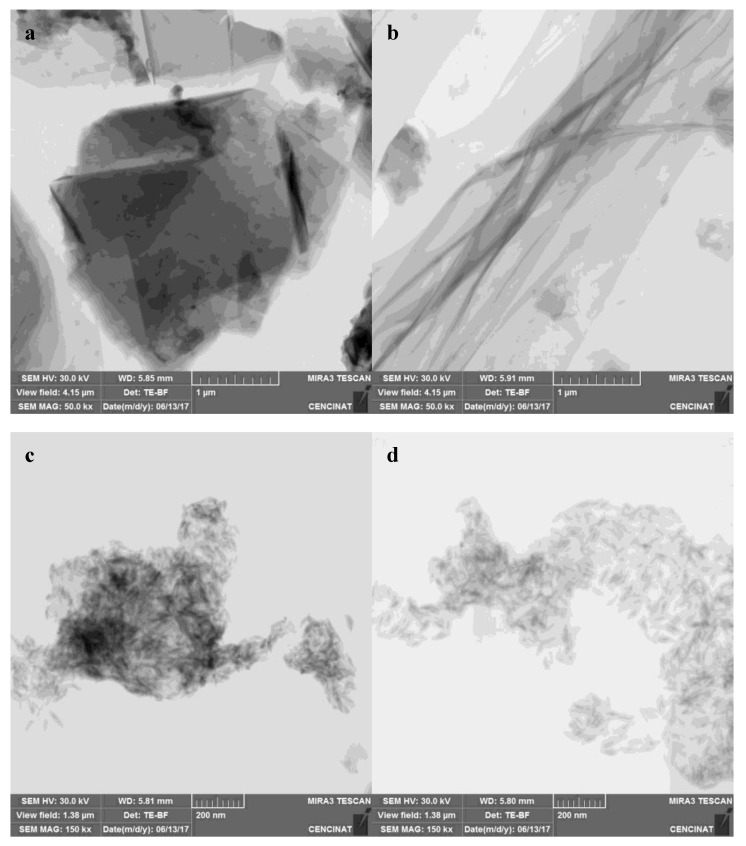
STEM images of (**a**) graphite, (**b**) GO, (**c**) α-FeOOH/rGO 1, and (**d**) α-FeOOH/rGO 2.

**Figure 6 nanomaterials-09-01580-f006:**
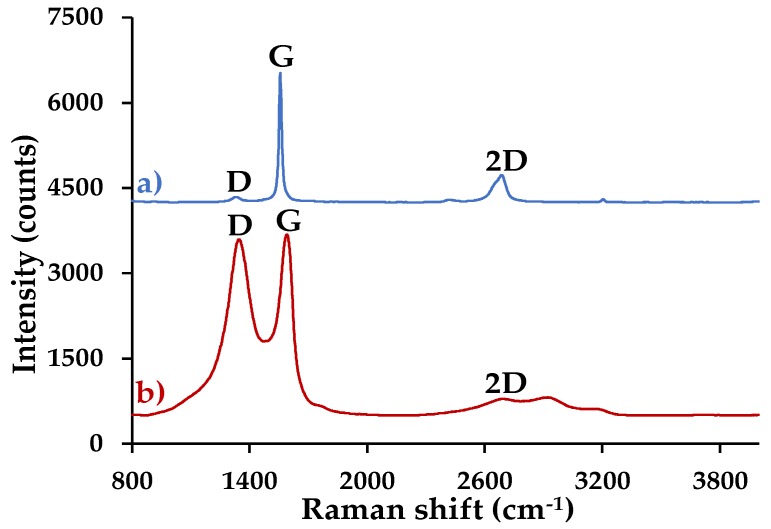
Raman spectrum of (**a**) graphite and (**b**) GO.

**Figure 7 nanomaterials-09-01580-f007:**
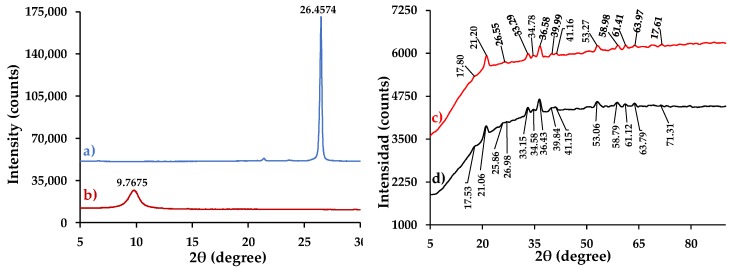
XRD patterns of (**a**) graphite, (**b**) GO, (**c**) α-FeOOH/rGO (composite 1), and (**d**) α-FeOOH/rGO (composite 2).

**Figure 8 nanomaterials-09-01580-f008:**
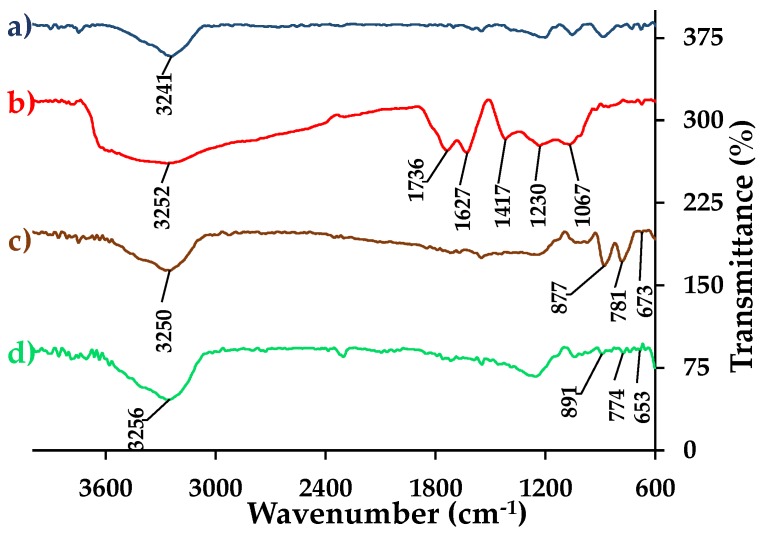
FTIR spectra of (**a**) GO reduced, (**b**) GO, (**c**) α-FeOOH/rGO 1, and (**d**) α-FeOOH/rGO 2.

**Figure 9 nanomaterials-09-01580-f009:**
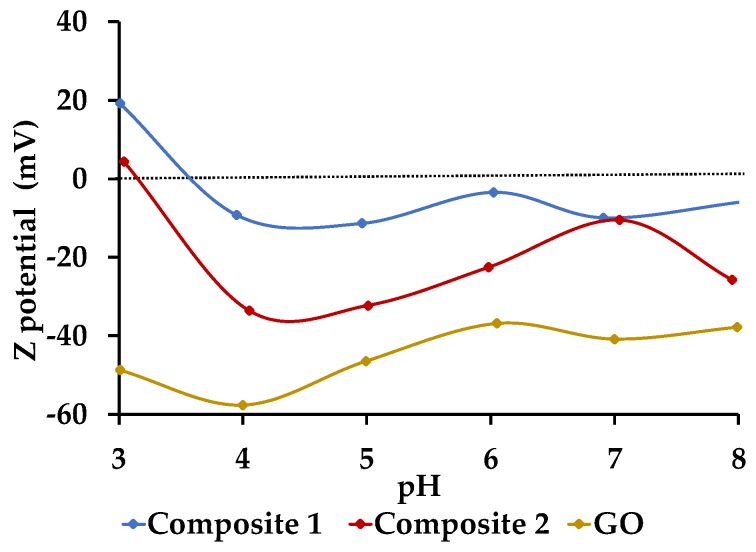
GO and composites surface charge in a pH range between 3 and 8.

**Figure 10 nanomaterials-09-01580-f010:**
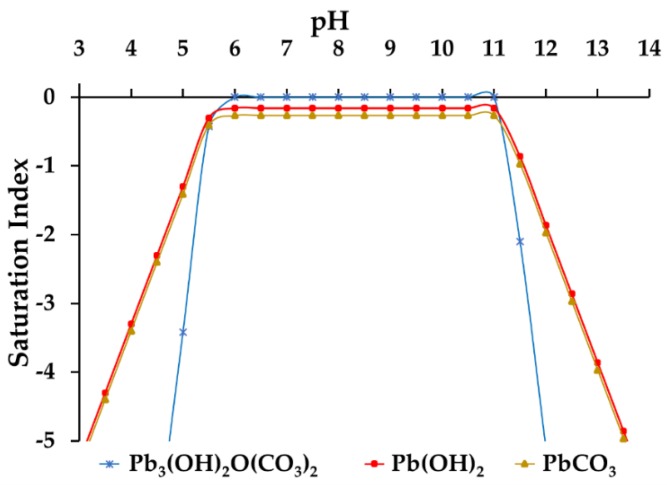
Distribution diagram of Pb(II) species as a function of pH. [Pb(II)]: 400 mg/L obtained with the MINEQL program.

**Figure 11 nanomaterials-09-01580-f011:**
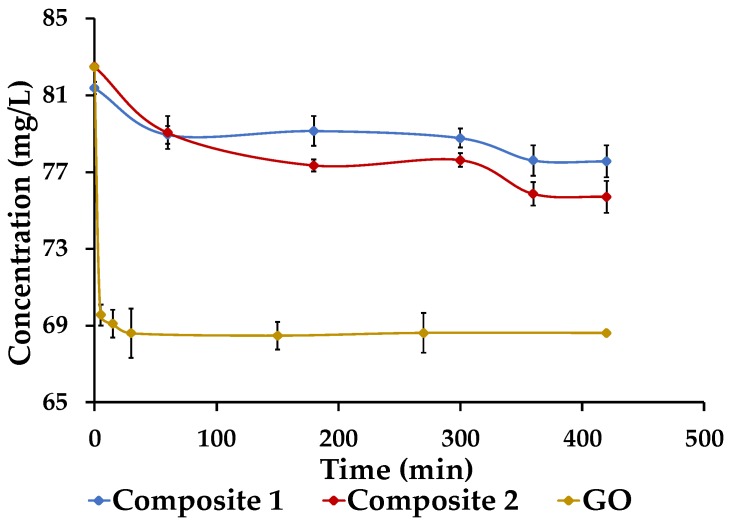
Adsorption kinetics for GO and the composites. [Pb(II)] = 80 mg/L; pH of the solution = 5; V = 50 mL; MSOLID = 5 mg; agitation = 250 rpm; T = 20 °C.

**Figure 12 nanomaterials-09-01580-f012:**
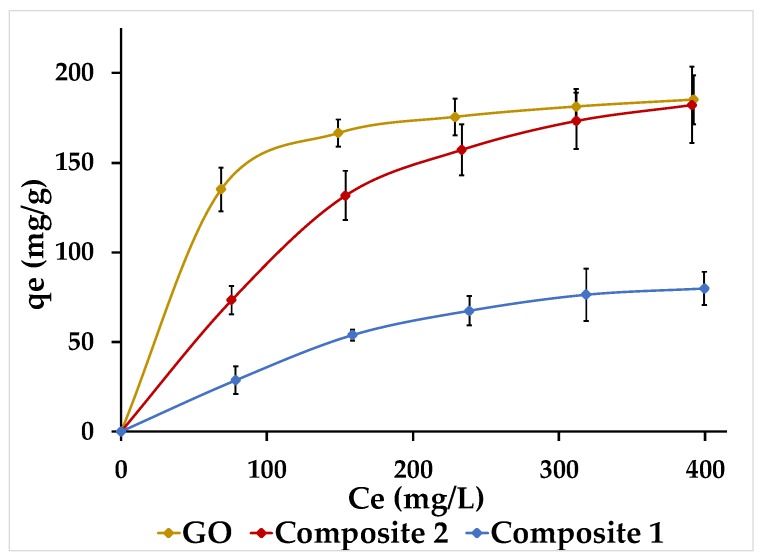
Isotherms for Pb(II) adsorption onto GO and composites; pH of the solution = 5; V = 50 mL; MSOLID = 5 mg; agitation = 250 rpm; T = 20 °C.

**Figure 13 nanomaterials-09-01580-f013:**
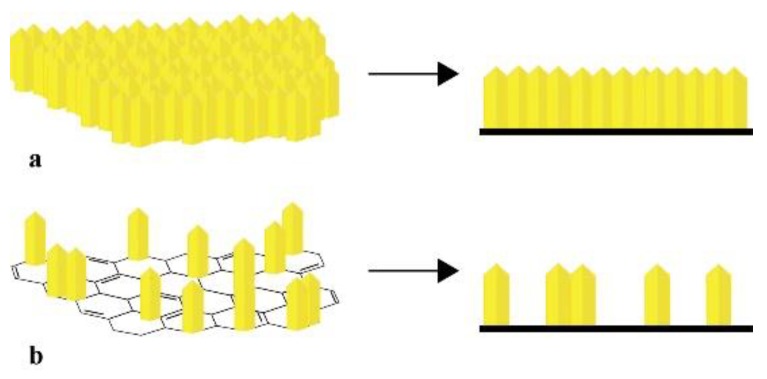
Illustration of α-FeOOH nanorods’ growth on the rGO surface: (**a**) α-FeOOH/rGO 1 and (**b**) α-FeOOH/rGO 2. Top view (left) and front view (right).

**Table 1 nanomaterials-09-01580-t001:** Normalized mass percent average value of graphite, GO, and composites measured by EDX.

Norm. Mass Percent (%)	C	O	Al	Si	S	Cl	K	Fe
Graphite	70.67	19.64	4.68	3.86	0.04	0.01	0.32	0.77
GO	45.02	38.87	4.83	3.94	3.72	2.03	0.50	1.09
Composite 1	21.16	33.56	0.56	0.52	0.76	0.03	0.03	43.37
Composite 2	16.55	38.17	0.42	0.29	0.84	0.01	0.02	43.70

**Table 2 nanomaterials-09-01580-t002:** Parameters associated with the Langmuir and Freundlich isotherms for the adsorption of Pb(II) in the synthesized compounds.

	Langmuir Isotherm			Freundlich Isotherm		
Material	q_max_ (mg/g)	K (L/mg)	R^2^	N	K (L/mg)	R^2^
GO	200	0.3	0.99	5.57	65.07	0.94
Composite 1	138.89	3.65 × 10^−3^	0.96	1.56	1.90	0.95
Composite 2	277.78	5.28 × 10^−3^	0.98	1.80	7.17	0.94

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
