# Peer review of "Applicability of Goethite/Reduced Graphene Oxide Nanocomposites to Remove Lead from Wastewater"

_nanomaterials, 2019, doi:10.3390/nano9111580_

Round 1
Reviewer 1 Report
The manuscript reports on the synthesis and the characterization of α-FeOOH/rGO composites. The suitability of such compounds to act as adsorbent for the removal of Pb(II) by wastewater through adsorption experiments was also assessed. The manuscript is well arranged scientifically and contains much new information. The paper is surely interesting and the obtained results are valid but an improvement of some sections should be provided. The paper contains useful information in the field and for these reasons, I consider the paper suitable for publication, but before this, some major corrections should be done. I suggest to improve the entire manuscript taking into account the following indications:
Entire manuscript:
The manuscript is easy to follow, but, anyway, I suggest a linguistic revision and a proper check for some grammar errors (for example at lines 24, 37, 176, 250, 322, 334). Some sentences need to be re-written: lines: 20, 56-57, 225-226, Avoid the improper use of the term graphene instead of GO (for examples at lines: 255, 288) The adsorption is “on” solid sorbent not “in” solid sorbent, please check the use of “in” and “on”
Introduction:
Add a reference for the sentence at lines 44-45 referred to the use of charcoal as adsorbent. Add details about the abundance of Pb(II) ions in wastewater, and the concentration considered not dangerous for human beings and animals. Re-write the aim of the work indicating clearly the number of composites produced and why they have been selected
Experimental section:
Which is the concentration of HCl reported at line 94? Section 2.5, use the plural form for Raman, XRD, and FTIR spectrum Why so high Pb(II) concentrations were explored? In this way no information about the performances of the materials with low concentration of the target ion (conditions more easily found in a real environment) were collected. Why the pure phase α-FeOOH was not produced and used as reference materials in the adsorption tests? The reference to literature data is possible only if the same experimental conditions were applied. Did the authors checked if nitrocellulose membranes catch Pb(II) ions during filtration processing? Remove the sentence "This same procedure was executed for the composites” at lines 146-147; it is a repetition of the previous sentence.
Results and Discussion section:
Subsection 3.1: the details about the effectiveness of graphite cleaning procedure are quite unnecessary, for this reason I suggest to remove the subsection 3.1 or to move it in the experimental section along the cleaning procedure details or to fuse it with the subsection 3.4.3. Subsection 3.2: what do the authors mean with the sentence at line 180: "the synthetized GO possesses domains of six to seven cyclic chains"? Subsection 3.2: please rephrase the sentence at line 184-186, it is too much speculative even if it is in accordance with previous literature findings. The authors do not check the textural properties of the samples (specific surface area, total pore volume, micro/meso porosity) and make some speculations on these properties of the synthetized materials only on the basis of previously reported data form other authors or only on the basis of SEM images. This approach is inappropriate and too much qualitative for a work about adsorbent performances evaluations. Subsection 3.4.1: The sentence at line 206-207 is too much speculative, rephrase it Subsection 3.4.1: Table 1, the content of Mn is not reported, does it mean that the GO sample is completely free of Mn?. Subsection 3.4.1: Line 211, change the word complex with composite and avoid possible nomenclature discrepancies along the manuscript text. Subsection 3.4.3: The description of Raman characterization is too simple, the typical parameter IG/ID is not provided, the material is a typical multi-layered structure and information about the number of layers is not clearly discernible. Subsection 3.4.3: at line 244 change “the layers of graphene” in “the graphene layers” Do the authors test the cyclability and the durability of their materials? A wide comparison with literature data on similar composite adsorbent properties should be helpful to understand the potential of the proposed materials. Subsection 3.7: the interpretation of Pb(II) adsorption data is too qualitative. The effect of agglomeration needs to be verify by additional experimental evidences.
Conclusions
Without a proper comparison with literature data the last sentence is quite exceeding, for this reason I suggest to rephrase or eliminate it.
Author Response
Dear reviewer, thank you very much for your comments and very accurate recommendations, below you can find the answers detailed of your observations, also in attached file you can find the manuscript with the observations included. Finally, I want to point out that, one more co-author was included in the revised manuscript, due to an involuntary error it was not included in the first submitted manuscript
Reviewer 1
Introduction:
“Add a reference for the sentence at lines 44-45 referred to the use of charcoal as adsorbent.”
Response: the reference in the manuscript was added
“Add details about the abundance of Pb(II) ions in wastewater, and the concentration
considered not dangerous for human beings and animals.”
Response: the requested details were included in the manuscript
“Re-write the aim of the work indicating clearly the number of composites produced and why they have been selected”
Response: the objectives were re-written in the manuscript, clearly indicating the number of composites.
Experimental section:
“Which is the concentration of HCl reported at line 94?”
Response: HCl concentration was added in the manuscript.
“Section 2.5, use the plural form for Raman, XRD, and FTIR spectrum”
Response: It’s done in the manuscript
“Why so high Pb(II) concentrations were explored? In this way no information about the performances of the materials with low concentration of the target ion (conditions more easily found in a real environment) were collected.”
Response: Lead concentrations proposed in this work, were focused on determining the maximum adsorption capacity, the adsorption model to which it fits and the model constants for the two composites in real samples of acid mine drainage, due our research group has focused on the study of these pollutants, but we are grateful for the observation, since once the model and constants have been found, it can be extrapolated to lower concentrations (around 10 mg /L).
“Why the pure phase α-FeOOH was not produced and used as reference materials in the adsorption tests?”
Response: Pb (II) adsorption process onto goethite was not reported in the manuscript because it is a subject widely studied previously. However, we have the data of the experimental part, the group decided not to include them in the document because we consider it would not provide new information.
“Did the authors checked if nitrocellulose membranes catch Pb(II) ions during filtration
processing?”
Response: Yes, we checked, the procedure is indicated in section 2.8 and we did not find any membranes effect.
“Remove the sentence "This same procedure was executed for the composites” at lines 146-147; it is a repetition of the previous sentence.”
Response: The changes were done.
Results and Discussion section:
“Subsection 3.1: the details about the effectiveness of graphite cleaning procedure are quite
unnecessary, for this reason I suggest to remove the subsection 3.1 or to move it in the
experimental section along the cleaning procedure details or to fuse it with the subsection
3.4.3.”
Response: Section 3.1. was reviewed and the details were omitted.
Subsection 3.2: what do the authors mean with the sentence at line 180: "the synthetized GO possesses domains cyclic chains"?
Response: It was corrected in the text.
Subsection 3.2: please rephrase the sentence at line 184-186, it is too much speculative even if it is in accordance with previous literature findings.
Response: It was reviewed and corrected.
The authors do not check the textural properties of the samples (specific surface area, total pore volume, micro/meso porosity) and make some speculations on these properties of the synthetized materials only on the basis of previously reported data form other authors or only on the basis of SEM images. This approach is inappropriate and too much qualitative for a work about adsorbent performances evaluations.
Response: We are grateful about the suggestion and we corrected the text to could be misunderstood, but we do not agree about this information too much qualitative. SEM and TEM images show the difference in the materials used and adsorption experiments confirm that behavior. This information could be complemented with BET analysis, but it was not in the project scope.
Subsection 3.4.1: The sentence at line 206-207 is too much speculative, rephrase it Subsection 3.4.1:
Response: It was done in the manuscript
Table 1, the content of Mn is not reported, does it mean that the GO sample is completely free of Mn?.
Response: According the mass balance, it does not exist a significative concentration of any other element.
Subsection 3.4.1: Line 211, change the word complex with composite and avoid possible nomenclature discrepancies along the manuscript text.
Response: It was corrected
Subsection 3.4.3: The description of Raman characterization is too simple, the typical parameter IG/ID is not provided, the material is a typical multi-layered structure and information about the number of layers is not clearly discernible.
Response: More details were added.
Subsection 3.4.3: at line 244 change “the layers of graphene” in “the graphene layers”
Response: It was done in the manuscript
Do the authors test the cyclability and the durability of their materials? A wide comparison with literature data on similar composite adsorbent properties should be helpful to understand the potential of the proposed materials.
Response: We did some experiments about the reuse of the material. At pH around 3.5, Pb(II) is desorbed, then it can be used again, but we still are working in this property. So we didn’t include it in this document.
Subsection 3.7: the interpretation of Pb(II) adsorption data is too qualitative. The effect of agglomeration needs to be verify by additional experimental evidences.
Response: We reviewed the interpretation and rewrite the misunderstood sentences.
Conclusions
- Without a proper comparison with literature data the last sentence is quite exceeding, for this reason I suggest to rephrase or eliminate it.
Response: It was corrected in the manuscript
Reviewer 2 Report
Dear authors, thank you, this is still very interesting and important topic,
The manuscript is readable, well arranged and offer a good results, congratulations. However I have 2 questions:
What about reversibility? Is it possible to adsorb such high amount of Pb only once? The materials characteristics, XRD, FTIR need corrections and improvements.
Summing, I can not find any drastic mistakes. Thus, in my opinion this work is worth to be published.
Author Response
Dear reviewer, thank you very much for your accurate observations, below you can find the detailed answers of your observations, also in attached file you can find the manuscript with the observations included. Finally, I want to point out that, one more co-author was included in the revised manuscript, due to an involuntary error it was not included in the first submitted manuscript
Reviewer 2
- What about reversibility? Is it possible to adsorb such high amount of Pb only once?
Response: We did some experiments about the reuse of the material. At pH around 3.5, Pb(II) is desorbed, then it can be used again, but we still are working in this property. So we didn’t include it in this document.
- The materials characteristics, XRD, FTIR need corrections and improvements.
Response: It was corrected in the manuscript.
Round 2
Reviewer 1 Report
The authors addressed most of the comments and questions I formulated during the first round of revision, so now in this form the manuscript is suitable for publication to me.
Reviewer 2 Report
Dear Authors, thank you, it looks ok, now